# Comparative Genomic Analyses of the Genus *Photobacterium* Illuminate Biosynthetic Gene Clusters Associated with Antagonism

**DOI:** 10.3390/ijms23179712

**Published:** 2022-08-26

**Authors:** Nyok-Sean Lau, Wooi Liang Heng, Noorizan Miswan, Nor Azura Azami, Go Furusawa

**Affiliations:** Centre for Chemical Biology, Universiti Sains Malaysia, Penang 11900, Malaysia

**Keywords:** *Photobacterium*, comparative genomics, pan-genome analysis, secondary metabolites, antagonism

## Abstract

The genus *Photobacterium* is known for its ecophysiological versatility encompassing free-living, symbiotic, and pathogenic lifestyles. *Photobacterium* sp. CCB-ST2H9 was isolated from estuarine sediment collected at Matang Mangrove, Malaysia. In this study, the genome of CCB-ST2H9 was sequenced, and the pan-genome of 37 *Photobacterium* strains was analysed. Phylogeny based on core genes showed that CCB-ST2H9 clustered with *P. galatheae*, forming a distinct clade with *P. halotolerans*, *P. salinisoli*, and *P. arenosum*. The core genome of *Photobacterium* was conserved in housekeeping functions, while the flexible genome was well represented by environmental genes related to energy production and carbohydrate metabolism. Genomic metrics including 16S rRNA sequence similarity, average nucleotide identity, and digital DNA–DNA hybridization values were below the cut-off for species delineation, implying that CCB-ST2H9 potentially represents a new species. Genome mining revealed that biosynthetic gene clusters (BGCs) involved in producing antimicrobial compounds such as holomycin in CCB-ST2H9 could contribute to the antagonistic potential. Furthermore, the EtOAc extract from the culture broth of CCB-ST2H9 exhibited antagonistic activity against *Vibrio* spp. Intriguingly, clustering based on BGCs profiles grouped *P. galatheae*, *P. halotolerans*, *P. salinisoli*, *P. arenosum*, and CCB-ST2H9 together in the heatmap by the presence of a large number of BGCs. These BGCs-rich *Photobacterium* strains represent great potential for bioactive secondary metabolites production and sources for novel compounds.

## 1. Introduction

*Photobacterium* (Vibrionaceae, Gammaproteobacteria) comprises a group of Gram-negative, facultative-aerobic, mostly halophilic, and motile bacteria [1]. The genus currently consists of 37 validly published named species (www.bacterio.net (accessed on 19 May 2022)), with *P. phosphoreum* as the type species [2]. All species of *Photobacterium* were originally thought to be luminescent. However, later studies found that only strains of *P. angustum*, *P. aquimaris*, *P. ganghwense*, *P. kishitanii*, *P. leiognathi*, *P. mandapamensis*, and *P. phosphoreum* display this characteristic [1,3]. Although members of this genus are ubiquitous in marine contexts, isolations of *Photobacterium* strains from non-marine habitats such as a saline lake, the rhizosphere of a terrestrial weed, and spoiled meat have been reported [4,5,6,7]. *Photobacterium* exhibits different lifestyles: some occur as free-living in seawater and sediments; others function as symbionts of the light organs of marine fish and squid, decomposers of dead fish, or pathogens for marine animals [1,8,9]. High diversity within *Photobacterium* has been reported, which could be linked to the lifestyles of the species [10,11]. *Photobacterium* has been studied for ecophysiological traits such as piezophilic, salt adaptation, bioluminescence, and motility [3,12,13].

*Photobacterium* also encompasses species that synthesize various secondary metabolites, including those with antibiotic activity [14]. Wietz et al. reported that a pyrrothine antibiotic, holomycin, was produced by *P. galatheae* S2753 (formerly *P. halotolerance*) [15]. Holomycin shows a broad antibiotic spectrum on several pathogenic bacteria belonging to the genera *Staphylococcus*, *Enterococcus*, *Escherichia,* and some marine bacteria such as *Roseobacter*, *Pseudoalteromonas,* and *Vibrio* [15,16]. For instance, *P. galatheae* S2753 also showed antagonism against a bacterial fish pathogen, *Vibrio anguillarum*, and a human pathogen, *Staphylococcus aureus* [17]. The mode of action of holomycin is speculated to inhibit RNA polymerase synthesis of the target bacteria [16]. In addition to the production of holomycin, *Photobacterium* sp. strain S2753 was also capable of producing two novel cyclodepsipeptides, solonamide A and B, which can inhibit the expression of the virulence gene of *S. aureus* [16]. These findings revealed that some *Photobacterium* strains have potential as biocontrol agents against pathogenic bacteria. However, the antagonism potential of *Photobacterium* spp. for use in aquaculture is still not well studied.

Increases in aquaculture production to meet global seafood demand have largely been accomplished through the intensification of current farming systems, resulting in a higher incidence of disease outbreaks. For instance, a shrimp disease, acute hepatopancreatic necrosis disease (AHPND), caused by *Vibrio parahaemolyticus* (VpAHPND), was reported in China in 2010 [18] and widely spread to South Asia from Latin America [19,20]. A detrimental reduction in shrimp production was caused by VpAHPND due to its high mortality rate, which is 100% on post-larvae of black tiger shrimp (*Penaeus monodon*) and Pacific white shrimp (*P. vannamei*), causing severe economic losses [18]. To control VpAHPND, several biocontrol strategies, such as probiotic supplementation, vaccination, and biofloc, have attracted attention in shrimp aquaculture [21,22,23]. However, these methods face several shortcomings, such as the requirement for long-term probiotic supplementation, the low efficacy of vaccination, and the difficulty of producing and maintaining biofloc in the aquaculture field. Therefore, alternative approaches for the control of VpAHPND have been demanded.

A pan-genome is the set of all non-redundant genes present in a given dataset, including the core, which is defined as the homologous genes present in all genomes in the dataset [24]. The status of the core- and pan-genomes is dependent on the number of analysed genomes, the species’ ability to acquire exogenous DNA via horizontal gene transfer, and its ecological niche [25]. A *Photobacterium* sp. (designated CCB-ST2H9) isolated from sediment collected in a previous metagenomic investigation [26] is the subject of this research. The aim of this study was to sequence the genome of CCB-ST2H9 to provide a glimpse into the genetic basis of antagonism. Core- and pan-genome analysis was performed to provide insights into the genome plasticity within *Photobacterium* and functional versatility for colonization of different niches. New clustering based on biosynthetic gene clusters (BGCs) profiles separated the *Photobacterium* into BGCs-rich and BGCs-low groups. Our study yielded a comprehensive understanding of the diversity of the biosynthetic gene clusters in *Photobacterium*, which will facilitate the use of the bacteria for bioactive metabolites production and as biocontrol agents in aquaculture.

## 2. Results and Discussion

### 2.1. Genomic Features

CCB-ST2H9 was isolated from estuarine sediment obtained from the Matang mangrove. Based on a 16S rRNA sequence similarity search performed using the EzTaxon database [27], CCB-ST2H9 is most closely related to *P. halotolerans* MACL01 (98.03%), *P. salinisoli* LAM9072 (97.65%), and *P. galatheae* S2753 (97.56%). These values are slightly below the species demarcation threshold of 98.7% [28].

The complete CCB-ST2H9 genome contained two chromosomes (Chr 1, 1.51 Mb; Chr 2, 1.51 Mb) and a plasmid (79.85Kb), with GC content of 50.3% (Table 1 and Appendix A). The GC content of CCB-ST2H9 was in the upper bound of *Photobacterium*, which varied between 38.6 and 50.9%. The CCB-ST2H9 genome falls in the *Photobacterium* genomes size range of 4.38 to 6.53 Mb, with the smallest genome observed for *P. malacitanum* CECT 9190, associated with diseased redbanded seabream, and the largest for *P. alginatilyticum* P03D4, isolated from a bottom seawater sample from the East China Sea [29,30]. Concomitantly, *P. malacitanum* and *P. alginatilyticum* annotated gene numbers were among the smallest and largest in *Photobacterium*, respectively. The CCB-ST2H9 genome has 99.2% gene space completeness following assessment with the BUSCO tool (Appendix A). Genome annotation of CCB-ST2H9 resulted in 4915 coding sequences, including 4703 protein-coding genes, 127 tRNA genes, and 37 RNA genes. A total of 3828 CCB-ST2H9 genes were assigned to 22 functional Clusters of Orthologous Groups (COG) classes, with transcription, amino acid transport and metabolism, and inorganic ion transport and metabolism being the most abundant pathways (excluding the unknown function category) (Appendix A). Gene Ontology (GO) annotation classified 3611 CCB-ST2H9 genes into three functional terms in which the top subgroups were catalytic activity, cellular process, and metabolic process (Appendix A). From the 2610 CCB-ST2H9 genes mapped to the Kyoto Encyclopedia of Genes and Genomes (KEGG), a significant proportion was involved in protein families: genetic information processing and environmental information processing, and protein families: signaling and cellular processes (Appendix A).

Genomic islands play an important role in bacterial genome evolution and adaptation as they are the probable horizontal origin of genes for pathogenicity, symbiosis, and metabolism [31]. A total of 13 genomic islands ranging in size from 10.2 to 35.8 Kb were detected in the CCB-ST2H9 genome (Appendix A). The genomic islands contain a considerable number of hypothetical genes, as well as genes related to energy, cell membrane, and signal transduction (Appendix A). In addition, five prophage-like elements were detected in CCB-ST2H9, one of which was classified as intact, three as incomplete, and one as questionable (Appendix A).

### 2.2. Phylogeny of Photobacterium

Although 16S rRNA phylogeny has been widely applied in species classification, the limitations of using this gene as a phylogenetic marker in Vibrionaceae have been noted previously [10,11]. Meanwhile, multigene phylogenies provide better resolution between phylogenetically close strains, and the tree topology is less affected by recombination events [32,33]. In this study, a phylogenomic analysis was performed based on the bacterial core genes from 37 *Photobacterium* strains, using *Vibrio cholerae* ATCC 14035 as an outgroup (Figure 1). This tree showed that CCB-ST2H9 clustered with *P. galatheae* S2753, forming a distinct clade with *P. halotolerans* DSM18316, *P. salinisoli* LAM9071, and *P. arenosum* CAU 1568, distant from the other *Photobacterium* strains. This grouping appears to be related to environmental sources. With the exception of *P. arenosum* from marine sediment, this clade consists of *Photobacterium* isolated from non-marine habitats: CCB-ST2H9 from estuarine sediment, *P. halotolerans* from a saline lake, and *P. salinisoli* from saline soil [7,34]. The remaining *Photobacterium* strains were grouped into the clades Damselae, Leiognathi, Ganghwense, Profundum, Phosphoreum, and Rosenbergii, consistent with prior reports [35,36]. While this clade classification can facilitate the study of large genera by grouping together lines of descent, it is not a standard in nomenclature [8].

The genome evolutionary relatedness between *Photobacterium* strains was depicted based on the average nucleotide identity (ANI) and digital DNA–DNA hybridization (dDDH). In agreement with phylogenomic clustering, the highest ANI was shared between CCB-ST2H9 and *P. galatheae* (88.95%), *P. halotolerans* (84.26%), *P. arenosum* (84.25%), and *P. salinisoli* (84.25%) (Figure 2). Accordingly, the highest dDDH percentage (32.7%) was between CCB-ST2H9 and *P. galatheae* (Appendix A). Both the sharing ANI and dDDH values between CCB-ST2H9 and other *Photobacterium* strains were below the proposed 95–96% and 70% cut-off of bacterial species definition [37,38]. Considering the 16S rRNA sequence similarity and ANI and dDDH results, strain CCB-ST2H9 is potentially a new species of the genus *Photobacterium*.

### 2.3. Core and Pan-Genome of Photobacterium

A pan-genome is comprised of core genes found in all strains, accessory genes (shell and cloud) common to two or more strains but not all, and unique genes present only in one strain. The orthologous clustering of protein-coding sequences from the 37 *Photobacterium* genomes analysed identified a total of 21,786 consensus gene clusters defining the pan-genome (Figure 3a). The size of the *Photobacterium* pan-genome would continue to expand with the increasing number of genomes incorporated in the analysis and can be considered open (Figure 3b). This pan-genomic feature is typical of taxonomic groups capable of colonizing diverse habitats and hence the opportunity to exchange genetic material with different sources [25,39].

More than half of the gene sets constituting the pan-genome (14,638 genes, 67%) belonged to the cloud cluster, whereas the shell and the soft-core clusters accounted for 26% (5587 genes) and 7% (1561 genes) of the pan-genome, respectively (Figure 3c). Both the cloud and shell clusters are subsets of the flexible genome that could be applied to infer an organism’s evolutionary trajectory and lifestyle or habitat adaptation [40]. These two clusters are hypothesized to have different rates of gene acquisition and deletion, with the cloud including rapidly gained and lost genes, and the shell containing slowly gained and lost genes [41]. The core genome presented in all examined *Photobacterium* strains had 894 genes, representing approximately 19% of the total genes in a genome, which indicates significant gene conservation among *Photobacterium*. The number of core genes decreased rapidly with the addition of genomes but stabilized after the addition of the 20th genome, suggesting that the core genome was closed (Figure 3d). In a previous *Photobacterium* pan-genome study, a higher number of core genes was observed, likely due to the inclusion of genomes from less diverse species in the analysis and the slightly smaller number of genomes in the dataset [10].

COG functional assignments revealed that the *Photobacterium* core genome was enriched in functions involving translation, coenzyme metabolism, and nucleotide metabolism and transport (Figure 4a). These core genes encode housekeeping functions related to fundamental processes in the cell that have been conserved in all species throughout evolution. In addition to translation and coenzyme metabolism, the soft-core genome was also abundant in transcription function. The abundant COG category transcription, which consists of transcriptional regulators, could enable *Photobacterium* to regulate metabolic processes, contributing to their adaptability to the local environment. Compared to the core genome, the *Photobacterium* flexible genome was well represented with functions related to energy production and conversion, cell wall/membrane/envelope biogenesis, and carbohydrate metabolism and transport. The flexible genome was enriched with environmental genes, such as those that allow *Photobacterium* to respond to environmental changes by utilising different types of carbon and energy sources for surviving under various conditions. The flexible genome contributes to functional versatility, which in turn improves the ecological success of the bacteria in diverse environmental niches. Analysis of CCB-ST2H9 specific genes revealed that only 34.8% (201 genes) were assignable to COG functions (Figure 4b). These unique genes that evolved in CCB-ST2H9 likely complement strain-specific activity via unknown mechanisms. Besides function unknown (31%), COGs associated with replication and repair, carbohydrate metabolism and transport, and cell wall/membrane/envelope biogenesis were most abundant among strain-specific genes.

### 2.4. Biosynthetic Potential of Photobacterium

To examine the potential of *Photobacterium* strains to produce bioactive metabolites, the genomes were screened for the presence of biosynthetic gene clusters (BGCs) using antiSMASH software. A total of 238 BGCs were detected amongst the 37 genomes, with beta-lactone (35), ribosomally synthesized and post-translationally modified peptide (RiPP)-like (33), and non-ribosomal peptide synthetase (NRPS) (30) being the most prevalent (Figure 5). Some BGCs co-localized as hybrid clusters, such as NRPS/trans-amino transferase polyketide synthase (transAT-PKS), NRPS/type I polyketide synthase (T1PKS), aryl polyene/ectoine, and NRPS/NRPS-like/T1PKS (Appendix A). Ten BGCs were identified in *Photobacterium* sp. CCB-ST2H9, including clusters potentially encoding beta-lactone, butyrolactone, cyanobactin, ectoine, NRPS, NRPS-like, NRPS/NRPS-like, RiPP-like, and siderophore (Table 2).

All *Photobacterium* except *P. lipolyticum* and *P. sanguinicancri* harbour a beta-lactone cluster. The CCB-ST2H9 beta-lactone cluster contained the biosynthetic gene encoding 2-isopropylmalate synthase, which is involved in the biosynthesis of beta-lactone [42] but shows no correlation with known clusters. CCB-ST2H9 also harbors a BGC for butyrolactone, a quorum sensing molecule related to the regulation of antibiotic production and morphogenesis [43]. The cyanobactin BGC uniquely found in the CCB-ST2H9 genome displayed no match to known clusters. Due to the diversity of gene products and cluster organization in BGCs, some predicted cluster regions do not resemble known database entries. *P. indicum*, *P. frigidiphilum*, *P. ganghwense*, *P. lipolyticum*, *P. galatheae*, *P. arenosum*, *P. haloterans*, *P. salinisoli*, and CCB-ST2H9 all have BGCs coding for ectoine that are reported to play a role in osmotolerance [44]. The lack of ectoine in other *Photobacterium* strains implies that they might utilize different strategies for osmoregulation in hypersaline environments. Although a RiPP-like BGC was detected in CCB-ST2H9, it did not show similarity to known clusters.

Many bacteria have developed the ability to secrete iron-chelating molecules or siderophores that bind iron to form siderophore-iron complexes that are then liberated internally in the cell [45]. Siderophores enable producers to scavenge dissolved iron from the environment and deprive competitors of it. Under iron-limiting conditions, siderophore-producing bacteria have a competitive advantage over other species that lack iron-chelating ability [46]. A siderophore BGC showing high homology (88%) to aerobactin from *Aliivibrio fisheri* ES114, which is a hydroxamate-type siderophore, was identified in CCB-ST2H9. The presence of the siderophore aerobactin in *P. halotolerans* MELD1 contributes to the protection of host plants from phytopathogens [47]. Similarly, the siderophore aerobactin in CCB-ST2H9 could contribute to its antagonistic potential. The identified NRPS in CCB-ST2H9 showing 100% similarity to BGC in the database include xenotetrapeptide of *Xenorhabdus nematophila* ATCC 19061. While *X. nematophila* has been studied for xenematide activity, the function of xenotetrapeptide in this strain has not been explored [48].

Another NRPS with potential biosynthetic novelty shows low homology (38%) with the holomycin gene cluster in *Streptomyces clavuligerus* ATCC 27064. Holomycin production has been reported in both *P. galatheae* and *P. halotolerans* and examined in terms of the biosynthetic gene cluster and physiological role [16,49,50]. Holomycin is a member of the dithiolopyrrolone class that displays antibiotic activity against a broad spectrum of bacteria and inhibits RNA synthesis [51,52]. Further sequence analysis revealed that the CCB-ST2H9 holomycin BGC contains 10 genes homologous to and arranged in the same manner as those in *P. galatheae* [49]. The cluster is formed by phosphopantothenoylcysteine decarboxylase, HlmF, metallophosphoesterase, HlmY, globin, HlmG, *N*-acyltransferase, HlmA, acyl-CoA dehydrogenase, HlmB, thioesterase, HlmC, flavin mononucleotide-dependent oxidoreductase, HlmD, MFS transporter, HlmH, transcriptional regulator, HlmX, and core protein NRPS synthetase, HlmE with a characteristic arrangement of cyclization (Cy), adenylation (A), and thiolation (T) domains (Figure 6). The largest block of conserved genes corresponding to *hlmA*-*hlmE* is conserved in all holomycin producers, including *S. clauvuligerus*, *Y. ruckeri*, *Pseudoalteromonas* sp., and *Photobacterium* [49,53,54,55]. The holomycin BGC was also found in the *P. arenosum* and *P. salinosoli* genomes, even though antibiotic production and antagonism in these species have not been investigated. The presence of nearly identical holomycin BGCs in *P. galatheae*, *P. halotolerans*, *P. arenosum*, *P. salinisoli,* and CCB-ST2H9 suggests that this cluster may have originated via a horizontal gene transfer (HGT) event from a donor closely related to these species. Overall, the annotation of CCB-ST2H9 genes involved in the biosynthesis of compounds with antibacterial and iron scavenging functions is in agreement with the strain’s ability to antagonize the growth of pathogenic bacteria.

High variability of the biosynthetic operons in terms of abundance and composition was observed across *Photobacterium*, with *P. galathea* carrying the highest number of BGCs (15), while *P. carnosum*, *P. damselae*, *P. iliopiscarium,* and *P. toruni* have the lowest (2). This is the first study to use clustering based on BGCs profiles to separate the *Photobacterium* into BGCs-rich and BGCs-low groups. Although no evidence linking environmental sources and BGCs was found, the frequency of BGCs types appears to correlate with genetic proximity. Notably, CCB-ST2H9 displays a similar frequency of BGCs with *P. halotolerans*, *P. arenosum*, *P. galathea,* and *P. salinisoli* that are phylogenetically close. Except for *P. alginatilyticum* and *P. proteolyticum* that are phylogenetically distant, these strains cluster together by the presence of a large number of BGCs in their genomes. Conversely, it could also be claimed that the acquisition of gene clusters leads to ecological diversity and speciation, forming a phylogenetically distinct clade of *P. arenosum*, *P. halotolerans*, *P. salinisoli*, *P. galathea,* and CCB-ST2H9. The wide distribution of bioactive metabolites in these *Photobacterium* strains suggests good potential for antibiotic production and as biocontrol agents.

In addition, the variable distribution of BGCs in *Photobacterium* is indicative of gene loss from the descendants of a cluster-harboring ancestor and HGT or recombination events. Genes encoding traits subjected to weak selection are more likely to be lost, whereas genes that confer positive benefits under certain conditions are likely to be gained. For instance, a polyunsaturated fatty acid (PUFA) biosynthetic cluster that is present in other *Photobacterium* strains was not detected in CCB-ST2H9. Since the PUFA cluster has been linked to cold temperature adaptation [56], it could have been lost in the evolution of non-psychrophiles *Photobacterium*. On the other hand, the presence of BGCs for antibiotics in *Photobacterium* such as *P. arenosum*, *P. halotolerans*, *P. salinisoli*, *P. galatheae,* and CCB-ST2H9 raises the intriguing question of the ecological significance of the BGCs to the producers. In habitats characterized by dense multispecies communities, bacteria compete with their neighbours for scarce resources [57]. Antibiotic compound productions are proposed as weapons that provide fitness advantage over other occupants of the same ecological niche [58]. Compared to free-living *Photobacterium*, densely colonized sediment and soil could be the factor driving the evolution of antibiotic-producing traits in strains such as *P. arenosum*, *P. salinisoli,* and CCB-ST2H9. This view is in line with results from pelagic ocean samples, which showed increased antagonism of particle-attached bacteria compared to free-living bacteria [59]. Overall, the distinct BGC profiles displayed by *Photobacterium* from diverse habitats provide clues that metabolite production capacity could be the result of habitat-specific adaptation.

### 2.5. Antimicrobial Activity of EtOAc Extract against Vibrio spp.

As analysed above, CCB-ST2H9 possesses the holomycin biosynthetic gene cluster in its genome. To confirm the production of antagonistic compounds, EtOAc extraction was performed from the culture supernatant. Disc diffusion assay of the crude extract resuspended in methanol was conducted against seven *Vibrio* spp., including VpAHPND, and two bacterial strains, *E. coli* DH5α and *Bacillus* sp. CCB-MMP212, as positive controls. Oliver and co-researchers reported that holomycin exhibited antimicrobial activity against both Gram-positive and negative bacteria such as *Staphylococcus* and *E. coli* [51]. Besides that, holomycin isolated from *P. galarheae* S2753 exhibited the antagonistic activity against several marine bacteria, including the genus *Vibrio* [49]. As shown in Figure 7, the crude extract also showed antimicrobial activity on *Bacillus* sp. CCB-MMP212 (Gram-positive) and *E. coli* DH5α (Gram-negative). All *Vibrio* spp. tested were susceptible to the crude extract (Figure 7). It was reported that AHPND could be caused not only by *V. parahaemolyticus* but also other non-pathogenic *Vibrio* species because of the introduction of the pVA1 plasmid carrying the *pirAB* gene encoding homologues of *Photorhabdus* insect-related (Pir) toxins [60]. This finding suggests that CCB-ST2H9 might have the potential as a novel biocontrol agent due to its broad host range of antagonistic compounds.

## 3. Materials and Methods

### 3.1. Bacterial Strains and Culture Conditions

*Photobacterium* sp. strain CCB-ST2H9 was isolated from estuarine sediment collected from the Matang Mangrove Forest in Malaysia (4°12′48.06″ N, 100°38′49.5204″ E) [26]. The isolation was performed on a high-nutrient artificial seawater agar medium (H-ASWM) (0.5% tryptone, 2.4% artificial seawater, 10 mM HEPES, pH 7.6) [61] following the method previously described [62]. The strain was deposited at the Microbial Biodiversity Library of the Centre for Chemical Biology (CCB-MBL), University Sains Malaysia. CCB-ST2H9 was cultured aerobically in H-ASWM at 30 °C. For antagonistic analysis of crude extracts from cell culture of ST2H9, the AHPND-causing *V. parahaemolyticus*-like strain (VpAHPND) was obtained from Universiti Malaysia Terengganu [63]. *V. harveyi* JCM 33361T, *V. owensii* JCM 16517T, and *V. proteolyticus* JCM 21193T were obtained from the Japan Collection of Microorganisms, whereas *V. alginolyticus* CCB-CTB329, *V. azureus* CCB-ST2H38, and *V. neocaledonicus* CCB-ST4H6 were from CCB-MBL. These strains were incubated on H-ASWM agar plates at 30 °C. The cells growing on the agar plate were inoculated into 10 mL of H-ASWM broth and incubated for 16 h at 30 °C with shaking (200 rpm). After the incubation period, 200 µL of the suspension was transferred to 10 mL of H-ASWM broth and incubated for 2 h at 30 °C with 200 rpm of shaking to obtain exponential phase cells. *Escherichia coli* DH5α (Invitrogen, Carlsabd, CA, USA) and *Bacillus* sp. CCB-MMP212 from CCB-MBL were used as positive controls and cultured on Lysogeny-Broth (LB) agar plate at 37 °C and Marine Agar (MB) plate at 30 °C, respectively.

### 3.2. Genome Sequencing

Genomic DNA was extracted from mid-logarithmic phase culture following the method of Sokolov et al. [64] with some modifications: sample lysis was performed with a buffer containing 50 mM NaCl, 50 mM Tris-HCl pH8, 50 mM EDTA, 2% SDS, and isopropanol precipitation was replaced by SPRI bead purification. The quality and quantity of isolated DNA were assayed using 1% (*w/v*) agarose gel electrophoresis and Qubit fluorimetry (Thermo Scientific, Waltham, MA, USA), respectively. The Illumina sequencing library was constructed using the NEBNext Ultra DNA kit (New England Biolabs, Ipswich, MA, USA) according to the manufacturer’s instructions. The genome of CCB-ST2H9 was sequenced on a NovaSeq 6000 (Illumina, San Diego, CA, USA) at 150 bp paired-end, generating 10.26 million reads totaling 1.53 Gb. The DNA library for Nanopore sequencing was prepared using the Ligation sequencing kit (SQK-LSK109, Oxford Nanopore, UK). The library was sequenced with a PromethION and yielded 182,476 reads with a total of 1.92 Gb.

### 3.3. Genome Assembly and Annotation 

Raw Illumina reads were trimmed with fastp (v0.23.2) (https://github.com/OpenGene/fastp, accessed on 12 August 2022) to remove low-quality bases and adaptor sequences. Nanopore sequences were preprocessed to remove reads with qscore below 7. Hybrid assembly of Illumina and Nanopore reads was performed using Unicycler (v0.5.0) (https://github.com/rrwick/Unicycler, accessed on 12 August 2022) [65] with default settings. Genome assembly metrics were computed with QUAST (v5.0.2) (http://quast.sourceforge.net, accessed on 12 August 2022). Assembly completeness was evaluated using BUSCO (v5.3.2) (https://busco.ezlab.org, accessed on 12 August 2022) against the bacteria_odb10 dataset. Genome annotation was carried out through Prokka pipeline (v1.14.6) (https://github.com/tseemann/prokka, accessed on 12 August 2022) [66] and the NCBI Prokaryotic Genome Annotation Pipeline (PGAP). Coding sequences (CDS) were functionally annotated via the eggNOG-mapper (v2.1.6) (http://eggnog-mapper.embl.de, accessed on 12 August 2022) and BlastKoala server (https://www.kegg.jp/blastkoala, accessed on 12 August 2022). Genomic Islands were detected by IslandViewer4 (https://www.pathogenomics.sfu.ca/islandviewer, accessed on 12 August 2022) with IslandPath-DIMOB and SIGI-HMM predictions, and prophage was screened using PHASTER (https://phaster.ca, accessed on 12 August 2022). Secondary metabolite gene clusters were predicted using antiSMASH (v6.0.1) (https://antismash.secondarymetabolites.org, accessed on 12 August 2022) [67] with detection strictness set to ‘relaxed’. The KnownClusterBlast feature of antiSMASH was used to compare identified clusters against known gene clusters from the Minimum Information about a Biosynthetic Gene Cluster (MIBiG) database. 

### 3.4. Comparative Genomics

The genome sequences of *Photobacterium* strains from different habitats were retrieved from the GenBank database (Appendix A). Genome-based phylogeny was carried out using the UBCG (v3.0) tool (https://www.ezbiocloud.net/tools/ubcg ) [68] based on 92 core genes. A maximum-likelihood tree was inferred with RAxML (v8.2.12) (https://github.com/stamatak/standard-RAxML, accessed on 12 August 2022) under the GTR + GAMMA model and 100 bootstrap replications. The phylogenetic tree was visualized using the Interactive Tree of Life (http://itol.embl.de (accessed on 12 August 2022)). The average nucleotide identity value was calculated using GET_HOMOLOGUES (v3.4.3) (https://github.com/eead-csic-compbio/get_homologues, accessed on 12 August 2022) [69] with parameters ‘-a CDS -A -t 0 -M’. The DNA–DNA hybridization was estimated in silico using the Genome-to-Genome Distance Calculator (v3.0) (https://ggdc.dsmz.de/ggdc.php, accessed on 12 August 2022) [70] with formula 2.

Gene clustering was performed with GET_HOMOLOGIES based on the bidirectional best-hit (BDBH), COGtriangles, and OrthoMCL algorithms. BLASTp searches were done with a 75% minimum alignment coverage and E-value set at 1×10^−5^. The core-genome was estimated from the consensus clusters defined by BDBH, COGtriangles, and OrthoMCL, whereas the pan-genome was computed from COGtriangles and OrthoMCL clusters.

### 3.5. Extraction of Antagonistic Compounds from CCB-ST2H9 and Its Disk Diffusion Assay

The extraction of antagonistic compounds from CCB-ST2H9 was conducted according to the method of Mansson [16] with slight modifications. CCB-ST2H9 was cultured in 300 mL of Sigma Sea Salt medium (4% Sigma Sea Salt, 0.4% glucose, and 0.3% casamino acids with 10 mM HEPES, pH 7.6) for 72 h at 30 °C with shaking (200 rpm). After the incubation, the cell suspension was centrifuged at 8000× g for 20 min to remove the cells. An equal volume of ethyl acetate (EtOAc) was added and mixed into the supernatant. The EtOAc fraction was evaporated using a rotary evaporator (Heidolph, Germany). The crude extract was resuspended in methanol (200 mg/200 µL), and the solution was subjected to a test for its antagonistic activity on *Vibrio* spp.

Double-layer agar plates containing test strains were prepared to test the antagonistic activity of crude extract. Four mL of H-ASWM soft agar (0.3%) containing 200 µL of the exponential phase cells of each *Vibrio* species were overlayed on H-ASWM agar plates. The disk soaked in 20 µL of the extract was placed on the center of the double-layer agar plate, and the plate was incubated for 1 day at 30 °C. The double-layer agar plates containing *E. coli* DH5α or *Bacillus* sp. CCB-MMP212 were prepared by LB agar and MB, respectively. The disk with soaked methanol was used as a negative control.

## 4. Conclusions

This study provides insights into the genomic characteristics of *Photobacterium* sp. CCB-ST2H9 and its antagonistic potential against *Vibrio* spp. *Photobacterium* has an open pan-genome encompassing a core genome encoding essential functions and a flexible genome containing environmental genes contributing to adaptation to diverse habitats. While phylogenomic analysis showed that CCB-ST2H9 was closely related to *P. galatheae*, *P. halotolerans*, *P. salinisoli,* and *P. arenosum*, genomic metrics including 16S rRNA sequence similarity, ANI, and dDDH values between CCB-ST2H9 and these species were below the boundary of a bacterial species. AntiSMASH prediction revealed that CCB-ST2H9 harbours 10 BGCs, including three NRPSs predicted to be involved in the production of antibiotics such as holomycin and xenotetrapeptide. These NRPSs, along with the BGCs for butyrolactone and siderophore, could contribute to the antagonistic potential of CCB-ST2H9. The BGCs profile revealed that CCB-ST2H9, along with the other six *Photobacterium* spp., formed a BGCs-rich group. In addition, the production of antagonistic compounds against the causative agent of shrimp disease was also experimentally confirmed. *Photobacterium* spp. are promising resources for the genetic exploration and exploitation of bioactive secondary metabolites.

## Figures and Tables

**Figure 1 ijms-23-09712-f001:**
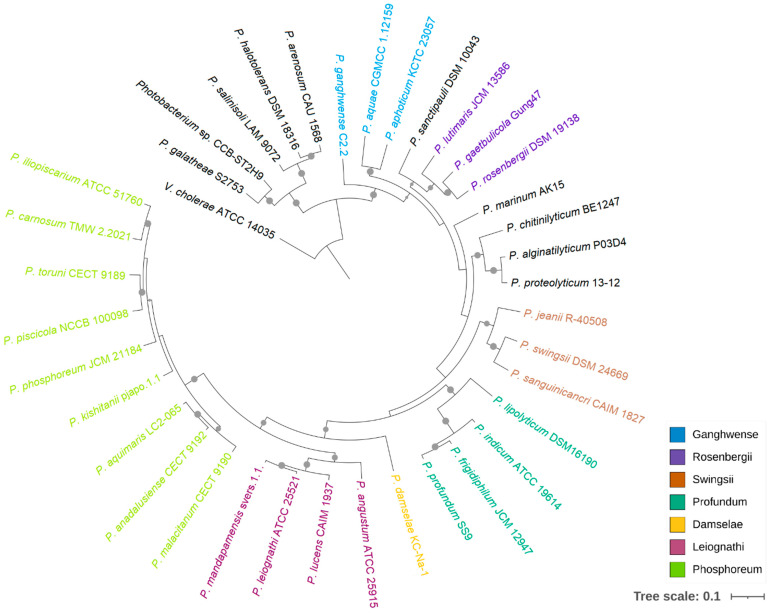
Phylogenetic relationship of *Photobacterium*. The maximum likelihood tree was constructed based on 92 bacterial core genes with UBCG. *Vibrio cholerae* ATCC 14035 was used as an outgroup, and support values of more than 50 are indicated by dots on nodes.

**Figure 2 ijms-23-09712-f002:**
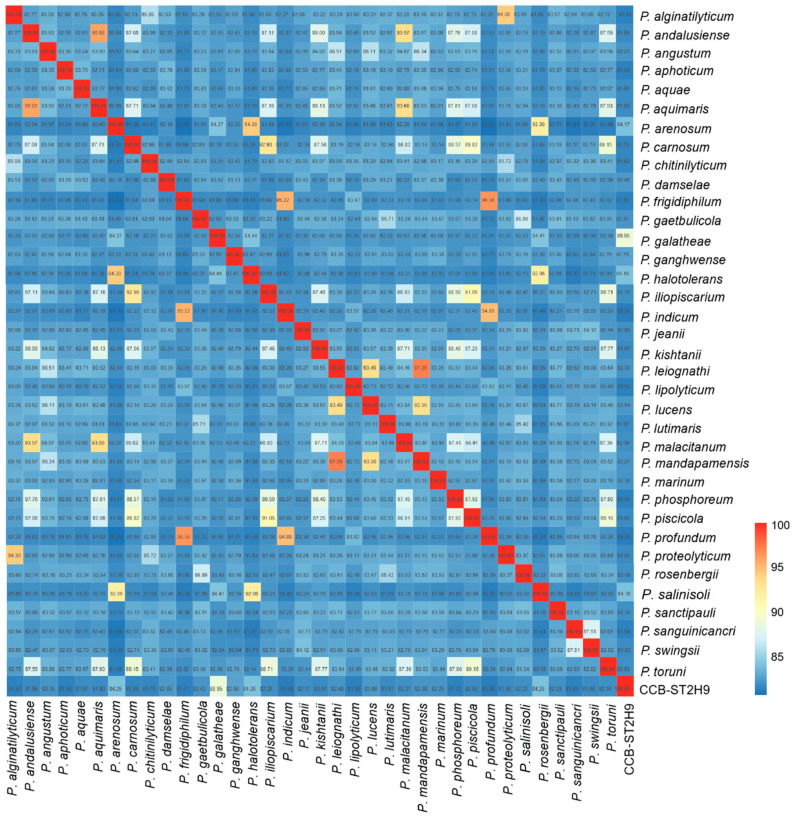
Average nucleotide identity (ANI) between *Photobacterium* genomes. The colour bar represents the value of ANI.

**Figure 3 ijms-23-09712-f003:**
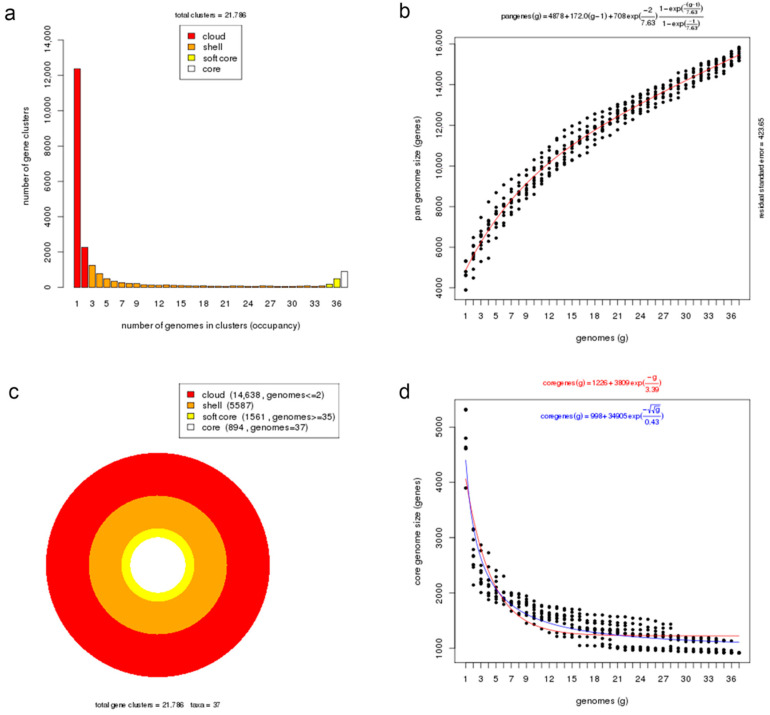
Core and pan-genome analyses of *Photobacterium* genus. (**a**) Barplot of the pan-genome matrix. (**b**) Plot of the estimated pan-genome size with Tellelin fit. (**c**) Cloud, shell, soft-core, and core clusters of the pan-genome. (**d**) Plot of the estimated core genome size with Tellelin and Willenbrock fit.

**Figure 4 ijms-23-09712-f004:**
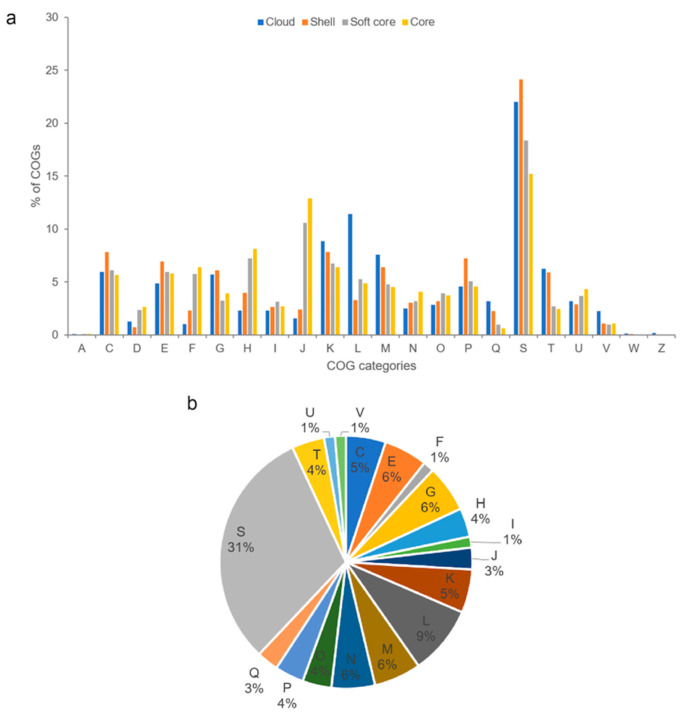
(**a**) Distribution of COG functional categories for cloud, shell, soft core, and core *Photobacterium* pan-genome clusters. (**b**) COG functional classification of strain CCB-ST2H9 specific genes. A: RNA processing and modification, C: energy production and conversion, D: cell cycle control and mitosis, E: amino acid metabolism and transport, F: nucleotide metabolism and transport, G: carbohydrate metabolism and transport, H: coenzyme metabolism, I: lipid metabolism, J: translation, K: transcription, L: replication and repair, M: cell wall/membrane/envelop biogenesis, N: cell motility, O: post-translational modification, protein turnover, chaperone functions, P: inorganic ion transport and metabolism, Q: secondary structure, S: function unknown, T: signal transduction, U: intracellular trafficking and secretion, V: defense mechanisms, W: extracellular structures, and Z: cytoskeleton.

**Figure 5 ijms-23-09712-f005:**
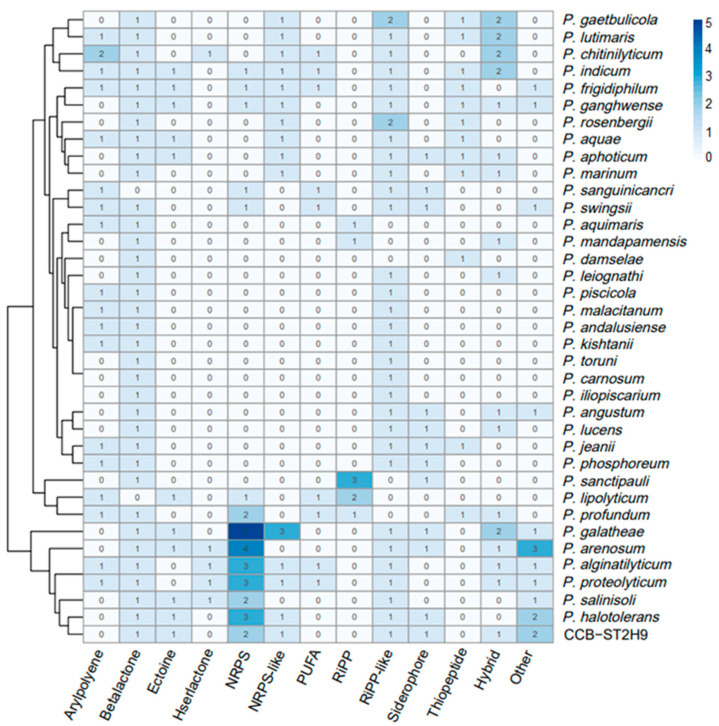
Heatmap representation of *Photobacterium* biosynthetic gene clusters (BGCs). The column dendrogram represents hierarchical clustering of bacteria by shared BGCs numbers, and the colour code represents the number of BGCs.

**Figure 6 ijms-23-09712-f006:**
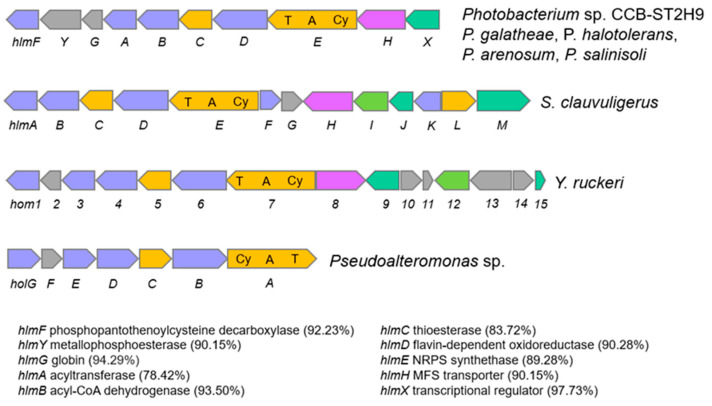
Comparison of holomycin biosynthetic gene cluster in *Photobacterium*, *S. clauvuligerus*, *Y. ruckeri,* and *Pseudoalteromonas* sp. Similarity (%) of genes in CCB-ST2H9 to those of *P. galatheae* are indicated in brackets. Light blue indicates oxidative gene. Orange indicates biosynthesis gene. Magenta indicates MFS transporter genes. Green indicated transcription regulator gene. Grey indicates gene with unknown function.

**Figure 7 ijms-23-09712-f007:**
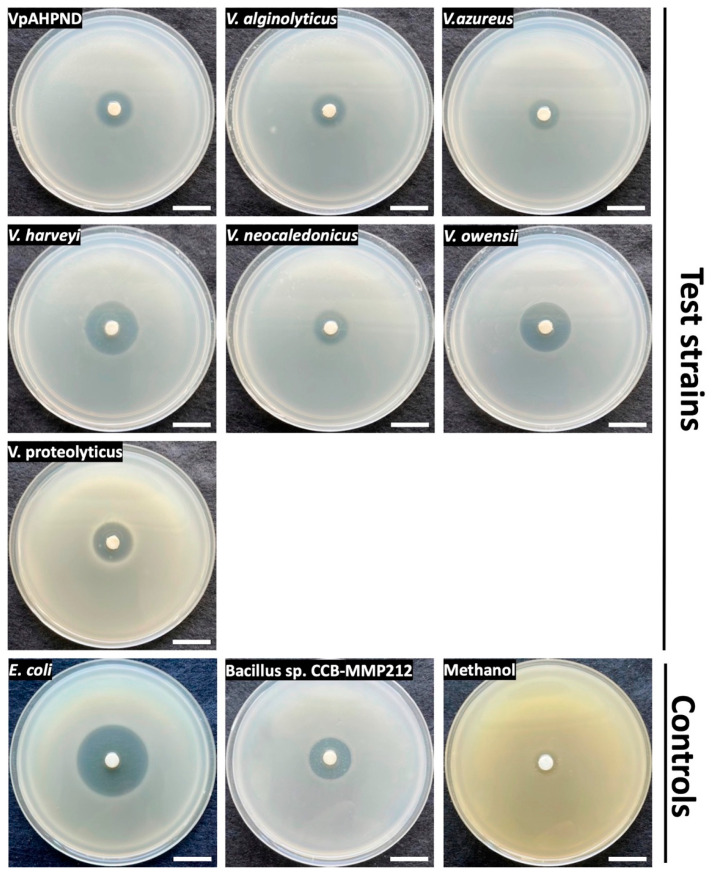
Disk diffusion assay of EtOAc extract on seven *Vibrio* species and control strains. Scale bars indicate 2 cm.

**Table 1 ijms-23-09712-t001:** Summary of *Photobacterium* sp. CCB-ST2H9 genome.

Features	CCB-ST2H9
Genome size (bp)	5,168,138
Number of contigs	3
G + C content (%)	50.34
Total genes	4915
Protein-coding genes	4703
Hypothetical proteins	2018
RNA genes	37
tRNA genes	127
Pseudogenes	173
Genes assigned to COG	3828
Genes assigned to GO	3564
Gene assigned to KEGG	2610

**Table 2 ijms-23-09712-t002:** Biosynthetic gene clusters in *Photobacterium* sp. CCB-ST2H9 predicted using antiSMASH.

Locus	Size (Kb)	Type	Most Similar Known Cluster	Similarity (%)
L4174_00195 − L4174_00375	39.8	NRPS-like	−	−
L4174_00650 − L4174_00955	85.9	NRPS/NRPS-like	Lipopolysaccharide	40
L4174_06195 − L4174_06225	10.4	Ectoine	Ectoine	66
L4174_07795 − L4174_07840	14.5	Siderophore	Aerobactin	88
L4174_08410 − L4174_08575	61.1	NRPS	Xenotetrapeptide	100
L4174_14130 − L4174_14230	27.0	Betalactone	−	−
L4174_15965 − L4174_16010	10.9	RiPP-like	−	−
L4174_20290 − L4174_20360	22.0	Cyanobactin	−	−
L4174_21690 − L4174_21850	43.4	NRPS	Holomycin	38
L4174_23295 − L4174_23355	10.8	Butyrolactone	−	−

−, no reference value.

## Data Availability

The *Photobacterium* sp. CCB-ST2H9 whole genome project is available in the NCBI database under the accession number CP100425-CP100427, SAMN25271433 (BioSample) and PRJNA800629 (BioProject).

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
