# Peer review of "Comparative Genomic Analyses of the Genus *Photobacterium* Illuminate Biosynthetic Gene Clusters Associated with Antagonism"

_ijms, 2022, doi:10.3390/ijms23179712_

Round 1

Reviewer 1 Report

This manuscript performed a comparative genomic analysis of the Genus Photobacterium for the detection of biosynthesis gene clusters. The authors analyzed 37 pan-genomes of the genus Photobacterium. The authors found a novel species in Photobacterium. Might be their final goals are to identify novel pathogens-antagonistic compounds for aquaculture or farming, for example, shrimp production. All genomic analysis is well performed using analyzing tool kits and finally, the novel Photobacterium species has functional BGCs (maybe holomycin-like) compound. However, there are not enough new insights for interest in the molecular science field. 

Reviewer 2 Report

In this manuscript, the authors have compared the species Photobacterium sp. CCB-ST2H9 with other Photobacterium genus.

1.         Authors have not mentioned the aim of the study. How it will be helpful for others.

2.         I think in the section Abstract, line 3 the word “CCB-ST2H9” may be removed.

3.         In Figure 1 are not clearly visible, kindly make it a circular representation using NJ Method

4.         Check the English language correction throughout the manuscript.

5.         Some grammatical mistakes are there kindly correct them.

 The Manuscript is well written. My decision is minor correction

Round 2

Reviewer 1 Report

The revised MS was well corrected in Grammatics. However, there are still not enough suitable data on BGCs products, especially holomycin, as antimicrobial substances. The present form of MS looks like a report for a bacterial new species. The author consumed many parts for general genome sequence data and identification of new species including supplementary data.

1.       Please changed to BGCs instead of biosynthetic gene clusters (BCGs) in the text all.

2.       The main data are antimicrobial activity through BGCs, maybe holomycin in the 3.5 section. Thus, this MS must be included in the comparison set with extract of P. galatheae for Vibrio susceptibility because P. galatheae already analyzed holomycin activity with a concentration in reference #49 or direct LC Mass data for exact holomycin functional evidence. I wonder holomycin concentration for Vibrio strains growth inhibition in extract because strain CCB-ST2H9 has many different types of BGCs in Table 2.

Round 3

Reviewer 1 Report

Thanks for your efforts. I already checked the author's revised version. They changed and corrected some sentences well. I hope the manuscript will be published in the current revision without the 3rd round.